# Devices and Methods for Dosimetry of Personalized Photodynamic Therapy of Tumors: A Review on Recent Trends

**DOI:** 10.3390/cancers16132484

**Published:** 2024-07-08

**Authors:** Polina Alekseeva, Vladimir Makarov, Kanamat Efendiev, Artem Shiryaev, Igor Reshetov, Victor Loschenov

**Affiliations:** 1Prokhorov General Physics Institute, Russian Academy of Sciences, 119991 Moscow, Russia; vi.makarov@physics.msu.ru (V.M.);; 2Department of Laser Micro-Nano and Biotechnologies, Institute of Engineering Physics for Biomedicine, National Research Nuclear University MEPhI, 115409 Moscow, Russia; 3Department of Oncology and Radiotherapy, Levshin Institute of Cluster Oncology, Sechenov First Moscow State Medical University, 119435 Moscow, Russia

**Keywords:** photodynamic therapy, photodynamic dosimetry, optical spectroscopy, fluorescence imaging, diffuse reflectance spectroscopy, fluorescent diagnostics, computer modeling

## Abstract

**Simple Summary:**

The current devices and methods for dosimetry of personalized photodynamic therapy of tumors have several disadvantages. For instance, they are unable to evaluate the entire tumor volume at great depth, and they lack the capacity to provide an accurate estimate of the effective light dose. Furthermore, they are limited by their ability to evaluate either only the photosensitizer concentration or only the hemoglobin oxygenation at a specific point on the tissue surface or the average value based on the finite tissue volume at little depth. Despite the advances in photodynamic therapy, the development of a precise and dependable methodology to ascertain the optimal light dosage for a given tumor remains a significant challenge. This is due to the pivotal role of light dosage in determining the efficacy of treatment outcomes.

**Abstract:**

**Significance:** Despite the widespread use of photodynamic therapy in clinical practice, there is a lack of personalized methods for assessing the sufficiency of photodynamic exposure on tumors, depending on tissue parameters that change during light irradiation. This can lead to different treatment results. **Aim:** The objective of this article was to conduct a comprehensive review of devices and methods employed for the implicit dosimetric monitoring of personalized photodynamic therapy for tumors. **Methods:** The review included 88 peer-reviewed research articles published between January 2010 and April 2024 that employed implicit monitoring methods, such as fluorescence imaging and diffuse reflectance spectroscopy. Additionally, it encompassed computer modeling methods that are most often and successfully used in preclinical and clinical practice to predict treatment outcomes. The Internet search engine Google Scholar and the Scopus database were used to search the literature for relevant articles. **Results:** The review analyzed and compared the results of 88 peer-reviewed research articles presenting various methods of implicit dosimetry during photodynamic therapy. The most prominent wavelengths for PDT are in the visible and near-infrared spectral range such as 405, 630, 660, and 690 nm. **Conclusions:** The problem of developing an accurate, reliable, and easily implemented dosimetry method for photodynamic therapy remains a current problem, since determining the effective light dose for a specific tumor is a decisive factor in achieving a positive treatment outcome.

## 1. Introduction

Photodynamic therapy (PDT) is one of the prospective methods for the treatment of precancerous and cancerous diseases, which already today shows high antitumor and antiviral efficacy. PDT works on the principle of light-induced activation of photosensitizers (PSs) and leads to cancer cell death mediated by reactive oxygen species (ROS), which ensures tumor growth cessation by triggering apoptosis and/or necrosis processes in the area of laser exposure, damage to blood vessels within the tumor that feed it and deliver oxygen, and has both local cytotoxic and systemic immunomodulatory effects [1,2,3,4,5,6,7,8,9,10,11,12,13,14,15,16].

However, despite the widespread use of PDT in clinical practice, there are no personalized methods for assessing the adequacy of tumor exposure to light depending on the tissue parameters that change during laser irradiation, which can lead to differently effective treatment outcomes. In the context of personalized PDT, the most important patient-specific parameters are backscattered radiation intensity, PS fluorescence intensity, tissue oxygen saturation, and blood filling.

The majority of clinical PDT procedures continue to employ a standard dose of the drug. Consequently, individual dosimetry has not yet been implemented on a widespread basis in clinical practice. This is primarily due to the discrepancy between the necessity of in vivo measurement and assessment of multiple parameters within the body to enhance treatment efficacy. It is of paramount importance to ensure that over- or under-irradiation of tissue is prevented when determining the optimal dosimetric parameter for each patient.

The presence of residual PS accumulation foci due to exposure to insufficient light dose is the main cause of recurrence after treatment, and excessive radiation dose can lead to superficial thermal damage to the tissue. The PDT dose delivered to different tumor sites may differ markedly due to the large variability in tissue parameters both between different patients and for a single patient [17,18]. 

Two approaches to calculating the adequacy of the delivered dose exist: explicit and implicit dosimetry. Explicit dosimetry is a method of direct measurement of all relevant factors affecting the effectiveness of PDT for a particular organ disease and a particular PS. This includes the estimation of parameters that define the limits of treatment efficacy. One of the primary objectives of precision dosimetry is to quantify parameters without the necessity for invasive medical procedures. Non-invasive measurement of tissue parameters is one of the main objectives of precision dosimetry. In the field of clinical cancer research, direct measurement is frequently employed, as some forms of cancer, such as skin cancer, cervical cancer, and head and neck cancer, are sufficiently accessible for direct observation.

The overarching objective of implicit dosimetry is to quantify all pertinent individual parameters, which, when combined, can yield an integrated parameter that is closely correlated with the therapeutic dose [19]. Illustrative examples include (i) light dose rate and fractionation, (ii) PS microlocalization in the vascular network in comparison to tumor parenchyma, (iii) localization effects of cell organelles, or (iv) the whole-body immune response to local PDT damage [20].

It is often exceedingly challenging to quantify and calculate all the variables that influence the delivered dose as a predictor of biological response. This review will focus on factors that can be measured intraoperatively in real time.

The depth of penetration and character of light propagation in the volume of biological tissue, and accordingly the effectiveness of PDT, are dependent upon the scattering and absorption coefficients, which vary according to the wavelength of laser radiation [21,22]. In other words, the propagation of light within the tissue is altered according to the change in optical properties that occurs during laser irradiation of the tissue [23]. The diffusion theory, when applied to a point light source, allows for the determination of the absorption and scattering coefficients of a specific tissue [24]. This, in turn, enables the calculation of the effective attenuation coefficient, which serves to characterize the depth of light penetration into the tissue under study. Due to the heterogeneous nature of biological tissue, the effective attenuation coefficient of radiation can vary considerably within the same tissue [24,25]. A change in optical properties can result in a significant change in the radiation flux density profile within the tissue. It is possible that the optical properties of the tissue may undergo change during the process of PDT [21,23,25]. For example, the scattering coefficient may increase after therapy due to changes in the concentration of red blood cells in the tissue caused by blood vessel damage. Blood is a complex system comprising various elements that contribute to its optical properties. These include red blood cells, platelets, leukocytes, their aggregates, and the surrounding medium (plasma). The disparity between the refractive indices of erythrocyte cytoplasm and blood plasma is the primary factor responsible for light scattering in blood. The greater the concentration of chromophores and scattering inhomogeneities in the tissue, the more optically dense the tissue is. This can lead to a significant increase (1.7–2.0 times) of the scattering coefficient in the spectral range from 600 to 1000 nm, resulting in a decrease in the depth of light penetration by 1.4–1.8 times [23]. The knowledge of the optical properties of the tissue enables the calculation of the dose of light delivered to a specific depth within the tissue [21,26]. 

An equally important factor is the presence and intensity of photochemical consumption of molecular oxygen by the cells of biological tissues to be treated by PDT. This affects not only the cytotoxic result of therapy, but also the depth of penetration of radiation, since oxy- and deoxyhemoglobin have different absorption spectra [27,28,29,30,31,32]. During PDT with high fluence rate, both photochemical oxygen consumption and microvascular thrombosis can deplete molecular oxygen, which may affect the treatment outcome. This is because tissue oxygenation may decrease to a level that is insufficient for further tissue destruction [33,34,35]. The change in tissue oxygenation during PDT is contingent upon the mode of light delivery, specifically whether it is continuous or pulsed irradiation with a high or low radiation fluence rate [36]. When photosensitized tissue is irradiated with a high fluence rate of radiation, the rate of molecular oxygen consumption during singlet oxygen generation may exceed the rate at which oxygen can be replenished by diffusion from the vascular network, resulting in a reduced photodynamic effect. Irradiation of photosensitized tissue with a low fluence rate of radiation increases the efficacy of PDT, but requires more time to provide effective treatment. This is due to the fact that a low fluence rate of radiation causes blood vessel damage, oxygen depletion, and changes in the depth of light penetration into the tissue. It should be noted that the process of damaging blood vessels, depleting oxygen, and changing the depth of light penetration into tissues takes longer than would be convenient for both medical personnel and the patient [36,37,38]. The fluence rate of radiation during PDT can be several times higher at a depth of 0.5–2 cm from the irradiated surface in well-oxygenated tissues compared to hypoxic tissues [39]. A significant increase (1.4–1.6 times) in the penetration of radiation with wavelengths of 630 and 650 nm is observed in well-oxygenated tissues, which is the most commonly used in PDT of various tumors [39,40]. For instance, in the context of decreased oxygenation of hemoglobin, the depth of light penetration at 630 nm was observed to decrease, whereas at 410 or 480 nm, it was observed to increase [39].

The final result of therapy is also influenced by the type of PS injected into the patient’s body and its microscopic and macroscopic distribution, the selectivity of accumulation of which affects the further effectiveness of fluorescence diagnostics and PDT. In the PD and treatment of tumors, drugs based on Ce6 and PpIX, which is a metabolite of 5-ALA, are widely regarded as the most effective agents in many countries. The accumulation of this substance is primarily observed in blood cells and vessels. During its circulation through the blood vessels, Ce6 demonstrates the capacity to penetrate tumor tissue with ease due to the higher permeability of tumor vessels compared to healthy tissue vessels [41]. Thus, PDT with Ce6 destroys the blood vessels feeding the tumor. These individuals are more susceptible to the formation of blood clots, a phenomenon known as thrombosis [42]. The cytotoxic effects of Ce6 result in the destruction of cells in a pathological state, through apoptosis and/or necrosis, and of tumor-associated macrophages, which contribute to the growth, invasion, and metastasis of the tumor [43]. This triple action mechanism has been demonstrated to significantly suppress tumor growth. 5-ALA induces PpIX, a fluorescent precursor of the heme biosynthesis pathway [44], in the mitochondria of cells of biological tissues. PpIX accumulates more intensely in tumor tissue cells than in normal cells [45]. In addition, PpIX can accumulate in macrophages [46]. It has been shown that ALA-PDT can damage pathological cells and change the metabolism of M0 macrophages towards M1 polarization, with M1 and M2 macrophages dying and being replaced by unpolarized cells [47]. The treatment result depends both on the specific photoproperties of the PS used, namely, the extinction coefficient and quantum yield of singlet oxygen, and on its distribution within the tissue/cell, which determines the site of action, which is predominantly vascular or cellular. 

This review considers devices and methods for implicit dosimetric control for personalized tumor PDT, the aim of which is to measure all relevant individual tumor tissue parameters during PDT that can yield an integrated parameter that closely correlates with the therapeutic dose [20]. Among all the proposed methods, optical imaging techniques, such as fluorescence imaging and diffuse light reflection spectroscopy, and computer modeling techniques are highlighted as the most frequently and successfully used in preclinical and clinical settings for treatment prediction.

## 2. Materials and Methods

A search was conducted using the Google Scholar internet search engine and the Scopus database to identify relevant research articles. The following keywords and search terms were employed in the query:

“photodynamic therapy” OR “pdt” OR “photodynamic”, AND

“optical spectroscopy” OR “fluorescence spectroscopy” OR “diffuse reflectance spectroscopy” OR “diffuse spectroscopy”, AND

“computer modeling” OR “monte carlo modeling”, AND

“dosimetry” OR “dose”

To enhance the precision of the search queries, general terms such as “cancer” or “tumor” were excluded from the search. The initial search yielded 253 peer-reviewed original research articles on the application of PDT in clinical practice. These articles were used to create a chart of the distribution of treated patients by localization from 1990 to 2024 worldwide (Figure 1; a table of publications can be found in the Appendix A).

Subsequently, 88 peer-reviewed research articles indexed in the Scopus database that applied implicit dosimetry control methods for personalized tumor PDT published between January 2010 and April 2024 were included in this review. It should be noted that review articles, conference proceedings, printed publications, dissertations, and reports were not included in the literature search strategy.

## 3. Results and Discussion

### 3.1. Optical Imaging

#### 3.1.1. Fluorescence Imaging 

The photodynamic exposure of a sensitized tumor results in the formation of ROS, which are responsible for tumor cell death. Additionally, the degradation of PS implies a decrease in its integrated absorption and fluorescence capacity. This phenomenon is commonly referred to as the photobleaching process and is often considered a dosimetric indicator of therapy efficacy, based on the assumption that the dose is proportional to the amount of PS consumed as a result of photobleaching during treatment [48]. Photobleaching is typically evaluated by the reduction in the fluorescence intensity of PS. There are highly accurate methods for in situ determination of PS concentration and photoproducts, but they remain too complex for clinical application. A more straightforward approach is to employ non-invasive fluorescence spectroscopy [49]. The real-time monitoring of PS fluorescence intensity and photobleaching can be employed as a dosimetric approach for the optimization and personalization of PDT parameters and the early prognosis of treatment outcomes [49,50].

A pilot clinical trial of interstitial PDT for inoperable recurrent glioblastoma was conducted in five patients. Oral administration of 5-aminolevulinic acid (5-ALA) at concentrations of 20 or 30 mg/kg was employed. Spectroscopic quantification of the accumulation and photobleaching of protoporphyrin IX (PpIX) during therapy was performed. The study involved the analysis of PS fluorescence intensity measurements and the stereotactic determination of the concentration of PpIX in tissue biopsy specimens. The correlation between fluorescence intensity and PpIX concentration and the clinical outcomes of treatment were analyzed [50].

A fluorescence imaging device that is capable of switching from white light reflection imaging (for the purpose of recording a color image) to fluorescence imaging (for the purpose of recording a fluorescent image) allows the PS photobleaching to adjust light doses during PDT. The effectiveness of this approach was demonstrated on 25 patients with actinic keratosis who were treated with topical application of 5-ALA-PDT methyl ester. There was no correlation between PS fluorescence intensity and lesion surface roughness, patient age, or pain, suggesting that these two parameters are not related to tissue production of PpIX. Nevertheless, the results demonstrated a robust linear correlation between fluorescence intensity and PS photobleaching, as well as treatment outcomes [51].

In a separate study, an optimized fluorescence dosimeter, comprising a spot probe and spectrometer, was utilized for dosimetry during PDT of actinic keratosis in 19 patients. Topical application of 5-ALA-PDT was employed to assess PpIX accumulation in superficial (405 nm blue wavelength excitation) and deeper (639 nm red wavelength excitation) tissue layers. The study demonstrated that the ratio of fluorescence under red and blue light excitation was sensitive to the depth-dependent distribution of PpIX. An elevated ratio of this parameter indicated a higher concentration of PpIX in deeper tissue layers, whereas a reduced ratio indicated a higher concentration of PpIX in superficial tissue layers. The ratio of PpIX fluorescence under excitation in the red and blue spectral ranges has the potential to be considered a dosimetric parameter of PDT [48].

Another research area that is currently experiencing significant growth is the development of phototheranostics methodologies based on the integration of PDT with diverse imaging technologies, enabling the concurrent delivery of both therapeutic and diagnostic procedures. This combination enables the heterogeneous distribution of PSs within tumor tissue to be accounted for through imaging, while simultaneously providing an effective and personalized therapeutic approach. The boundaries of cervical neoplasms in PDT were initially delineated by video fluorescence imaging and subsequently refined by spectral fluorescence measurements (Figure 2) [52].

The efficiency of exposure, and consequently, the PDT dose were evaluated by monitoring the change in Ce6 photobleaching during the treatment process. This approach enabled the PDT process to be monitored at all stages of treatment. At Ce6 concentrations of 0.8–1.2 mg/kg, fluorescence images of the cervix exhibited well-defined contrast between pathologic and normal surrounding tissue, allowing for the assessment of PS distribution over the entire measured area with a resolution of 1 mm [13,14,52]. In a separate study, fluorescence diagnostics and 5-ALA- and hexyl aminolevulinate-PDT (HAL-PDT) were employed to examine cervical dysplasia and vulvar leukoplakia in both continuous and pulsed laser irradiation modes. The fluorescence intensity of PpIX in the cervical epithelium was found to be five times higher following HAL application compared to that observed following 5-ALA application. The administration of 5-ALA and HAL did not result in an increased fluorescence intensity in vulvar tissues complicated with leukoplakia [53].

A comparable methodology and fluorescence imaging device for the detection of tumors and their spatial localization within the patient’s body, as well as the assessment of PS accumulation and photobleaching during PDT, were evaluated in a patient with recurrent multifocal basal cell carcinoma of the right parotid region at a concentration of the introduced PS Photoditazine (VETA-GRAND LLC, Russia, Moscow) of 1 mg/kg. The device demonstrated the kinetics of PS photobleaching and the recovery of fluorescence intensity following laser irradiation. Clinical studies on tumors of other patients demonstrated significant differences in PS photobleaching kinetics, which underscores the importance of this parameter for the optimization and personalization of therapy [54].

In one study, the light dose was estimated by measuring the light power (in watts, W) of a light probe balloon filled with a 21-milliliter solution of Intralipid 0.01%. The light power was converted to light flux density values (in W/cm^2^) for further analysis. A digital image was also compared to the results. This methodology permitted the establishment of the theoretical illumination profile of the developed light probe, which is a component of an intraoperative dosimetry system designed to enhance light delivery during intrapleural PDT [55].

One of the most cost-effective devices may be a fluorescence scanner comprising a smartphone with a dedicated imaging application, a 3D-printed backbone to standardize measurements, an emission filter to enhance PpIX fluorescence, and an LED ring, emitting light at 405 nm. Excitation of PpIX fluorescence demonstrated high performance in lesion visualization and quantification of PpIX accumulation and photobleaching in lesion foci of three patients with actinic keratosis following topical application of a 20% 5-ALA solution (Figure 3) [56].

A controller including a light source that performed two functions, PS activation for therapeutic purposes (production of ROS) and PS excitation for spectrofluorimetric measurements, was tested on patients. The ability to track the reference trajectory of PS photobleaching in real time by modulating the width of light pulses during treatment sessions was demonstrated. The study demonstrated that the proposed method markedly reduced the variability in PS photobleaching dynamics, both between patients and within the same patient. This real-time trajectory tracking principle can be applied to any other parameter characterizing the effectiveness of therapy in addition to PS photobleaching, as it is easily compatible with devices used in PDT, such as lasers and spectrometers [49].

It is well established that the time integral of the product of PS concentration, as determined by measuring PS fluorescence in vivo, and light fluence is referred to as the PDT dose and represents a crucial dosimetric quantity for PDT. However, the measured PS fluorescence may be distorted due to variations in the absorption and scattering of both excitation and fluorescence light in the tissue. It is possible that variations in tissue optical properties may be misinterpreted as differences in PS concentration. The dose of irradiation delivered to tissues can vary significantly between different patients and within the same patient. The quantification of PS concentration necessitates the correction of the measured fluorescence intensity with a correction factor accounting for changes in optical properties [57].

Four-channel and eight-channel PDT dosimetry systems for the simultaneous acquisition of light dosimetry and PS fluorescence data were developed and tested on eight patients with pleural mesothelioma [18,58]. Prior to the intraoperative photodynamic therapy (iPDT), patients were administered Photofrin at a dose of 2 mg/kg as an intravenous infusion. PDT was conducted using a dye laser with a 632 nm wavelength. The total fluence was 60 J/cm^2^. The empirical correction function of Photofrin fluorescence was determined empirically using Monte Carlo modeling for a number of physiologically significant optical properties of tissues. The parameters of the correction function were determined experimentally using phantoms that mimic the optical properties of biological tissue. In vivo measurements of Photofrin fluorescence demonstrated minimal photobleaching during PDT, yet substantial heterogeneity in Photofrin concentration was observed within and between patients. The irradiation doses delivered to 22 sites in the pleural cavity in eight patients exhibited a 2.9-fold difference within patients and an 8.3-fold difference between patients, with the same administered Photofrin concentration and tissue irradiation parameters. The distribution of light flux density was monitored using an IR navigation system, which permitted the generation of two-dimensional maps of light flux density over the entire pleural cavity, rather than just the study sites. The dose of light delivered to tumors during PDT can serve as a useful prognostic indicator of treatment outcome, as it accounts for differences in tissue optical properties and PS concentration distribution both between different patients and within the same patient [18,57].

The method of near-infrared tumor phototheranostics, which employs the separate use of PpIX and Ce6, permits the control of PS distribution in the near-infrared range and the assessment of PS photobleaching during PDT. The phototheranostics procedure included the use of a single laser in the red spectral range to diagnose the distribution of PS and to perform PDT. Near-infrared phototheranostics using PpIX and Ce6 were conducted on optical phantoms and tumors of patients with oral leukoplakia and basal cell carcinoma. The fluorescence of PpIX and Ce6 was recorded in the near-infrared region. The photobleaching of PpIX and Ce6 was evaluated during PDT by monitoring the change in PS fluorescence, which enabled the personalization of the duration of photodynamic treatment for deeper tumors (Figure 4) [16].

A significant number of PDT dosimetry methods fail to consider the three-dimensional heterogeneity of tumor tissue and the impact of patient movement during therapy on dose distribution. This can result in disparate treatment efficacy. The PDT of tumors was controlled using 3D image analysis and dynamic irradiation planning, as well as monitoring of light flux density when targeting heterogeneous 3D lesions by controlling the brightness of the LED array [59,60]. The findings indicated that the adjusted model of tumor illumination by an LED matrix with optimal brightness distribution according to the size and shape of the lesion enhances the uniformity of tissue irradiation and the efficacy of therapy, even when the patient moves during PDT. The model enables the calculation of the light dose at any point of the moving lesion, thereby demonstrating a personalized approach to tumor treatment [59,60].

#### 3.1.2. Diffuse Reflectance Spectroscopy

The efficacy of PDT is contingent upon the oxygen saturation of blood vessels, which can be influenced by tissue perfusion [61]. Tumor tissues are typically characterized by severe hypoxia, which significantly reduces the effectiveness of PDT. Furthermore, the oxygen consumption that occurs during PDT can exacerbate tumor hypoxia, which may result in a number of adverse effects, including neoangiogenesis, tumor invasiveness, and tumor metastasis [62]. Moreover, damage to blood vessels during PDT can result in alterations to tissue perfusion. These changes can be utilized as biomarkers to monitor microvascular responses to PDT. Diffuse light reflection spectroscopy represents one of the most popular methods for non-invasive monitoring of microvascular tissue parameters during PDT, including hemoglobin oxygenation, blood volume fraction, and tissue perfusion. This method allows for the prediction of therapy efficacy, as the cytotoxic effect of PDT depends in part on the concentration of molecular oxygen in the tissue [63]. This was demonstrated on 15 patients with malignant pleural mesothelioma undergoing macroscopic complete resection followed by iPDT mediated by Photofrin. During this procedure, light distribution was recorded using isotropic detectors with equal probability of light collection from all sides, and hemoglobin oxygenation in tissues was analyzed by examining diffuse light reflection spectra. Moderate oxygen depletion was observed in tissue areas with high hemoglobin oxygenation before PDT, which may indicate the presence of a treatment effect [64].

It was demonstrated that it is feasible to monitor alterations in microvascular parameters of wine stain foci in patients before and during PDT by diffuse reflectance spectra in the 500–600 nm range using diffuse reflectance spectroscopy. Studies have demonstrated that wine stains exhibit stronger absorption than normal skin in the spectral range studied. This may be related to the abundance of blood vessels in the lesion areas. Significant differences were observed in the microvascular parameters of wine spots between patients and within patients prior to PDT. In all patients, the mean values of hemoglobin oxygenation, blood volume fraction, and perfusion exhibited a significant increase following the initiation of PDT, and subsequently, demonstrated a gradual decline [65,66,67].

The analysis of diffuse backscattered reflectance spectra was employed during PDT of periodontal tissues with methylene blue. This approach permitted non-invasive monitoring of soft tissues of the oral cavity in 15 patients with various forms of cerebral palsy, including spastic diplegia and atonic–astatic form. Additionally, it was utilized in conjunction with gingivitis to identify areas of tissues with low hemoglobin oxygenation levels, thereby enabling precision photodynamic treatment. The authors demonstrated a significant increase in hemoglobin oxygenation from 50 to 67% (*p* < 0.001) and a notable reduction in blood volume in the microcirculatory bed of periodontal tissues 12 days after therapy (Figure 5) [68].

In a study by Quintanar L. et al., a system was developed to adjust the parameters of superficial 5-ALA-PDT before and during treatment. This system is equipped with a 630 nm high-brightness light-emitting diode with automatic regulation of therapeutic power, taking into account the model of light distribution over depth. Additionally, it includes a module for measuring tissue oxygen saturation based on two LEDs with wavelengths of 660 nm and 940 nm [69]. The authors posit that the system meets the requisite standards for clinical and preclinical PDT studies and can be utilized for a range of PDT modalities with minimal modifications.

The efficacy of PDT is contingent upon the interplay between the absorbed light dose, which is influenced by the therapeutic laser power and the optical properties (absorption and scattering) of the biotissue, and the PS concentration [19]. If the optical properties of the tissue are known, direct modeling can be employed to determine the light dose distribution within the tissue. This may facilitate the development of individualized treatment plans for patients and ensure the delivery of an effective light dose to the entire lesion area. To this end, the authors developed and validated a system to determine the optical properties of tissue and the concentration of methylene blue in it from measured diffuse reflectance spectra of abscess cavities of 13 patients before PDT (Figure 6) [70].

A correlation was identified between tissue oxygen saturation and the measured concentration of methylene blue in the tissue. This phenomenon can be explained by the local oxidation of hemoglobin to methemoglobin, which is unable to bind oxygen [71] at high concentrations of methylene blue [70,72,73,74]. A further study included 25 patients with pleural malignancies, whose tissues were examined by diffuse reflectance spectroscopy before and after PDT mediated by 2-(1-hexyloxyethyl)-2-devinylpyropheophorbide-a. The data were subjected to analysis using a nonlinearly constrained multi-wavelength and multi-variable algorithm in order to determine the tissue optical properties, tissue oxygen saturation, and total hemoglobin concentration [75].

The inverse problem of determining the absorption coefficient by diffuse reflectance spectroscopy methods is often a time-consuming task that differs for each particular patient. To overcome these difficulties, the use of artificial neural networks that determine the absorption coefficient using a single wavelength has been proposed. Fredriksson I. et al. presented an ANN-based method that directly outputs hemoglobin oxygen saturation and hemoglobin concentration using the shape of measured diffuse reflectance spectra as input [76]. The ANN was trained on spectra generated from a three-layer tissue model with hemoglobin oxygen saturation and hemoglobin concentration. Once the ANN has been trained, the computations are extremely rapid, taking only microseconds for each set of spectra. This can be contrasted with the previous nonlinear optimization method, which required approximately 0.2 s if the appropriate starting point was known (e.g., the solution of the previous time point), and on the order of 1 min if a global search had to be employed to sufficiently reduce the risk of local-optimal solutions. The results observed in patients during shoulder occlusion demonstrated the expected pattern. Specifically, oxygen saturation decreased to zero during the occlusion phase, while hyperperfusion resulted in oxygen saturation levels that were well above baseline at the time of release.

#### 3.1.3. Combined Method of Fluorescence Imaging and Diffuse Reflectance Spectroscopy

The influence of structural differences between similar tumors in different patients, as well as differences between primary and metastatic (or secondary) tumors on PS and molecular oxygen distribution, makes it challenging to select optimal irradiation parameters during PDT [77]. Consequently, researchers proposed a method of simultaneous monitoring of PS photobleaching, hemoglobin oxygenation, and tumor blood flow, as well as tumor optical parameters during PDT. This method allows for the characterization of the irradiated sensitized tissue’s state without interrupting the therapeutic light exposure to the tumor. This approach enables the control of the light dose during therapy, personalization of the exposure parameters for the tumor, and an increase in the effectiveness of treatment. The selectivity of Ce6 accumulation was found to affect the incidence of vascular thrombosis. The intensity of backscattered therapeutic radiation is a useful indicator of vascular thrombosis during Ce6-PDT. Figure 7 shows the results of combined spectroscopic monitoring of tumors after initial and repeated PDT [78].

Patients who initially received PDT exhibited a high degree of selectivity in Ce6 accumulation (Figure 7a). In contrast, patients who repeatedly received PDT demonstrated a reduced level of selectivity in Ce6 accumulation compared to normal tissues (Figure 7b). When the Ce6 accumulation in the tumor is high (Figure 7c), the intensity of diffuse-scattered laser radiation during PDT at a certain time point begins to increase, while the PS photobleaching begins to decrease. This phenomenon may be attributed to the complete thrombosis of blood vessels in the area of treatment and increasing tissue density. In the event of low Ce6 accumulation in the tumor, the hemoglobin oxygenation at the initial stage of laser exposure remains unchanged (Figure 7d). Consequently, the intensity of diffuse-scattered laser radiation and blood perfusion decrease, which may be associated with temporary vasoconstriction [78].

In preclinical models, monitoring hemoglobin oxygenation and PS fluorescence has been demonstrated to predict the outcomes of PDT. Hemoglobin oxygenation, blood volume fraction, and PS fluorescence in superficial lesions of patients’ head and neck were measured during PDT. It was demonstrated that an increase in the oxygenation of lesions prior to PDT is associated with a positive outcome of therapy. The simultaneous monitoring of hemoglobin oxygenation and PS fluorescence in tissues allows for the personalization of PDT by determining the parameters of the lesions based on the measured characteristics of these lesions [79,80].

The non-invasive quantitative control of blood flow, blood volume, blood oxygen saturation, and PS concentration in the tumor has led to an improvement in the therapy of head and neck cancer in patients. Both between patients and within the same patient, there were differences in the values of the investigated parameters before and after treatment. A correlation has been demonstrated between higher blood flow and hemoglobin oxygenation in the tissue during PDT and increased efficacy of therapy. A reduction in PS concentration (photobleaching) was observed as an additional indicator of effective PDT [81].

One investigation evaluated the potential of using diffuse reflectance spectroscopy to non-invasively assess hemoglobin oxygen saturation and total hemoglobin concentration in precancerous or superficial microinvasive oral lesions in patients simultaneously with fluorescence spectroscopy to evaluate PS accumulation in tissues (Figure 8) [82].

Furthermore, a distinct system for non-contact tissue fluorescence spectroscopy was employed, which facilitated continuous assessment of PS photobleaching throughout the course of PDT. The study revealed notable discrepancies in optical properties, PS concentration, and hemoglobin oxygenation between patients. The degree of hemoglobin oxygenation was found to be dependent on the depth of the study. The degree of hemoglobin oxygenation was greater in the superficial layers of the lesion than in the deeper layers [82].

A further study demonstrated that the monitoring of PS fluorescence and hemoglobin oxygenation in tissues during low-dose PDT enabled a preliminary local increase in PS concentration in tumor tissues [83]. This study demonstrated that low-dose radiation can reduce pain during PDT. The efficacy of low-dose PDT in clinical settings may be attributed to the reduced vascular damage caused by the low radiation fluence rate employed. Conversely, a high fluence rate can markedly reduce the oxygen concentration in tumor tissues, which may result in treatment ineffectiveness [84,85]. Furthermore, low-dose PDT possesses the advantage of allowing for determination of the optimal energy dose required to achieve the desired therapeutic effect, based on changes in PS fluorescence intensity and hemoglobin oxygenation observed during therapy (Figure 9) [83].

A novel approach to layer-by-layer laser dosimetry in biological tissues, which enables the precise selection of an individual therapeutic dose during laser irradiation, was developed through the real-time monitoring of radiation fluence, PS concentration, and tissue hemoglobin oxygenation at different depths. Furthermore, the light absorption efficiency of these chromophores in tissue layers was evaluated, as well as the morphological changes in irradiated tissue, including the volume content of blood vessels and the ratio of different forms of hemoglobin. Initially, diffuse reflectance spectra of the tissue were obtained by measuring them with a fiber optic spectrophotometer, at a fixed distance from the area where excitation light was delivered [86]. The mathematical apparatus utilized to transform the acquired data into a form suitable for analysis included approximating functions for the diffusely reflected light fluxes [87,88], as well as a method for rapidly calculating the light flux distribution over the depth of a multilayer optically dense medium [87].

Another approach to optimizing PDT treatment parameters on tumor tissue, including spot diameter and radiation fluence, was implemented through numerical simulation of laser radiation propagation with varying fluence and spot diameter in cervical tissue and the monitoring of PS photobleaching and hemoglobin oxygenation degree on phantoms (optical models) imitating the optical properties of cervical tumor tissue [88]. Based on an analysis of all the data obtained and considering the depth of the cervical lesion established through histological examination, the most appropriate laser radiation energy parameters for PDT (spot diameter and radiation fluence) for cervical dysplasia and cancer in 107 patients were determined, thus minimizing side effects and preventing disease progression and recurrence [13,14,52].

### 3.2. Computer Simulation

When planning PDT of tumor tissues, it is of the utmost importance to be aware of the optical properties of each specific tissue in order to correctly assess the optimal dose of light required to achieve a therapeutic effect. The fluence rate of radiation in tissue is dependent on its optical properties, which exhibit significant variability between patients and within the same patient. A high scattering coefficient and a low absorption coefficient of biological tissue can result in the propagation of a significant portion of radiation in the opposite direction and diffuse reflection from the tissue–air interface back into the tissue due to the difference in the refractive indices of biological tissue (n = 1.41) and air (n = 1), which, in turn, can cause an increase in the radiation fluence rate on the tissue surface and influence the PDT result [89]. Analytical and numerical methods for calculating the light penetration into tissue can assist in the planning process.

There are several methods for quantifying the spatial distribution of light in biological tissue. These include numerical Monte Carlo simulations and analytical approaches based on the light transfer equation and diffusion approximation. Analytical theory, which is based on Maxwell’s equations, is the most fundamental approach. However, there are difficulties associated with obtaining accurate analytical solutions [90,91]. The transport equation that models the penetration of light into biological tissue does not have an analytical solution. Consequently, the Monte Carlo method is typically employed to determine the propagation of photons in tissue [92].

Monte Carlo simulation is regarded as the gold standard in biophotonic modeling, enabling the consideration of all relevant physical and optical properties of biological tissues, although this does entail a significant computational demand. This tetrahedral mesh modeling method is particularly attractive due to its capacity to refine the mesh as desired, thus accommodating complex geometries or user-specified resolution requirements. The Monte Carlo method is a valuable tool for general evaluation and planning of PDT, offering significant cost savings compared to other predictive methods. A complete system for assessing the prospective energy dose in interstitial photodynamic therapy (iPDT) is demonstrated using Monte Carlo simulations to generate dose–volume histograms based on tetrahedral mesh geometry. The estimation of dose–volume histograms has been demonstrated to be an effective variance reduction scheme, which significantly reduces the number of required packages, and therefore, the execution time required to achieve an acceptable level of reliability of results [93,94].

It is crucial to recognize that the design and method of laser exposure, in addition to the geometry of the irradiated organ, may influence the energy profiles of light fields in certain instances. Specifically, the potential of methylene blue-mediated PDT for the treatment of infected abscesses with intracavitary delivery of methylene blue was investigated using computed tomography data from a representative population of abscess patients and Monte Carlo simulations of light delivery. The study demonstrated that in abscess cavities or hollow organs such as the bladder, the light fluence is approximately two to six times higher than the light fluence calculated per tissue surface area. The increase in light fluence rate, known as the “integrating sphere effect”, is a consequence of light transmitted into the tissue being diffusely reflected back into the hollow space, where further propagation and interaction with the tissue can then occur. At higher light absorption by tissues, for example, when tissues are irradiated at shorter wavelengths, the “integrating sphere effect” and the associated increase in light fluence rate are significantly reduced. The concentration of methylene blue can significantly influence the magnitude of the “integrating sphere effect”, and thus, should be monitored during research [95].

Another interesting phenomenon is the scattering properties of tissues. The numerical modeling of the propagation of laser radiation in biological tissue revealed that an increase in the fluence rate of laser radiation in the near-surface layer of the cervical tissue model relative to the fluence rate of the incident radiation was accompanied by an increase in the spot diameter in the range from 0.2 to 15 mm. This enabled the parameters of PDT exposure on real cervical tissues sufficient for effective irradiation of the tissue along the invasion depth without causing superficial thermal damage to the tissue to be established. Figure 10 shows the dependence of the relative fluence rate on the penetration depth (distance at which light intensity decreases by *e* times) for different spot diameters in the cervical tissue model, obtained as a result of Monte Carlo simulation [88].

The figure also demonstrates the outcomes of an experimental investigation conducted with spot diameters of 5, 10, and 15 mm, taking into account the alteration in the fluence rate within the cervical tissue model relative to the incident fluence rate, which are indicated by colored markers. As illustrated in the figures, at a constant incident radiation fluence rate, an increase in the spot diameter results in a corresponding increase in the radiation penetration depth. It was demonstrated that an expansion in the spot diameter from 0.2 to 15 mm led to a rise in the radiation fluence rate in the near-surface layer of the cervical tissue relative to the incident fluence rate [88].

A priori knowledge of the optical properties of organ tissue and the maximum allowable threshold photodynamic dose is an essential basis for the development of personalized PDT protocols, both under normal conditions and using organ perfusion techniques such as lung perfusion. An improvement in light penetration into tissues and the enlargement of tumor treatment volumes in PDT of lung metastases, in particular in diffuse disease, can be achieved by the replacement of blood with a low-cellular-content perfusate. A quantitative evaluation of the optical properties of the two perfusates and the photodynamic threshold for 5-ALA and Ce6 is presented. The perfused lungs of pigs and humans were placed in an ex vivo lung perfusion system and perfused with cell-free solution or blood. Isotropic diffusers were placed in the bronchi and on the lung surface for light transmission measurements, from which absorption and light-scattering properties at several wavelengths were calculated. In a separate experiment, pigs were administered with 5-ALA or Ce6 and irradiated with lung tissue at increasing doses. The low-cell perfusate significantly reduced the tissue absorption coefficient, increasing light penetration depth by 3.3 mm and treatment volumes by 3 times. The photodynamic threshold for lung treatment with 5-ALA was consistent with other malignancies. Ce6 levels were below the detection limit in lung tissue and did not demonstrate photodynamic-induced necrosis [96].

The correct choice of light source for PDT ensures optimal light delivery and achievement of tumor necrosis. The photodynamic dose delivered to the tumor during PDT was modeled using a three-dimensional Monte Carlo radiation transfer method for non-laser and laser light sources at varying beam diameter and tumor position in the tissue, with light doses up to 75 J/cm^2^. The final photodynamic dose delivered to a tumor depth of 2 mm using the Paterson light source was 2.75, 2.50, and 1.04 times greater than that delivered by the Waldmann 1200, Photocure, and Aktilite light sources, respectively. Tumor necrosis occurred at a depth of 2.23 mm and increased to 3.81 mm for wavelengths of 405 and 630 nm, respectively. Increasing the beam diameter from 10 to 50 mm had little effect on the depth of necrosis. The depth of necrosis decreased as the tumor moved deeper into the tissue. When the tumor moved to a depth of 1.6 mm from the tissue surface, the depth of necrosis decreased by 93% [97,98].

Radiation monitoring systems for PDT of cancer of hollow organs, such as the bladder, provide high selectivity over the entire organ wall. This selectivity is achieved by taking into account the parameters that determine treatment efficacy, in particular, the average optical properties of the mucosal layer, which determine the fluence rate, as well as the shape and position of the emitter in relation to the organ wall. The efficacy of three irradiation monitoring systems was evaluated in a preclinical model to determine their ability to provide selective bladder PDT given previously determined photodynamic threshold values for bladder, mucosal, and urothelial cancers. Additionally, a specific PS absorption coefficient was considered. Monte Carlo simulations of light propagation were performed for six human bladders during therapy to determine a range of tissue optical properties. Monte Carlo simulations demonstrated that irradiation monitoring systems should include at least three sensors. It is important to minimize light scattering within the bladder cavity to prevent increased radiation inhomogeneity. Surface dose values vary significantly with bladder shape and volume, but are less dependent on the optical properties of the tissue [99].

In order to plan the most effective delivery of radiation to the tumor, two three-dimensional light distribution models were developed. The first of these was a simple empirical model that directly calculates the radiation fluence rate at the surface of the nasopharyngeal cavity using a simple linear function that includes the scattering contribution as a function of the light source at a distance from the detector/cavity surface. Secondly, an analytical model based on Lambert’s law was developed. The input variables of this model are the diffuse reflection constant of the medium and the total output power of the spherical diffuser. The diffuse reflection constant determines the fraction of incident light re-emitted by the surface back into the cavity, while the total output power of the spherical diffuser allows for the calculation of the fluence rate on the surface of the optical phantom. The models were evaluated using three 3D-printed optical phantoms that simulated the optical properties of tissue and one porcine tissue phantom. The predicted radiation fluence rates of both models were accurate to within ±20%, allowing for the determination of the optimal source location and light source output power settings [100].

Monte Carlo modeling is based on the probabilistic nature of the interaction of photons with biotissue. When a photon, passing through the tissue, reaches the interaction site, it can either be scattered or absorbed. The probability of different events is determined by the optical properties (scattering and absorption coefficients) of the medium in which the photon is traveling. Therefore, the tumor structure strongly influences the spatial distribution of therapeutic light in the tissue. Monte Carlo simulations of radiation transport were employed for the first time to investigate the effects of three-dimensional tumor structures on light fluence rate, photodynamic doses, and PpIX fluorescence emitted not only from the tumor surface but also from below the surface. In addition, the effects of inhomogeneities within the tissue were investigated by incorporating models of fractal cluster structures. The incorporation of three-dimensional fractal structures into the theoretical study resulted in a deeper penetration of therapeutic radiation and a higher photodynamic dose. In one of the investigated three-dimensional fractal structures, an increase in effective penetration depth of 1 mm was observed in comparison to the equivalent homogeneous model [101].

A method for estimating the fluence rate at depth in a seven-layer skin model that takes into account the optical properties of the tissue has been presented. This method is based on the Monte Carlo method for multiple light sources during PpIX PDT. This approach enabled the duration of treatment to be controlled, thereby improving the effectiveness of therapy. The normalized PpIX absorption coefficient was used to determine both the effective fluence rate at the skin surface and at depth in the tissue. Subsequently, the total effective fluence rate and the in vitro concentration of PpIX in the tissue were estimated at different incubation and treatment durations. The product of the threshold fluence rate and threshold PpIX concentration resulted in a photodynamic dose used as a cytotoxicity threshold. Finally, reference tables for determining the minimum time required for treatment, depending on the light source and lesion depth, were presented [102].

New algorithms that generate personalized, high-quality treatment plans by optimizing the position of light sources and their power, while minimizing damage to organs at risk during tumor removal, are improving the effectiveness of PDT. An algorithm for optimizing treatment plans that models the physics of light propagation using Monte Carlo simulations and determines the number of photon emitters, their position and input power, taking into account tissue anatomy and response to treatment, has reduced the risk of organ damage by an average of 46% compared to manual placement of emitters commonly used in clinical practice. Different sources of perturbation that may give better results have been proposed and their effects have been studied. To minimize the number of steps taken (and effective execution time) while maintaining the quality of the result, a method based on machine learning with reinforcement was employed to determine which perturbation strategy to perform at each iteration. Simulations were conducted on virtual brain tumors, simulating real cases of glioblastoma multiforme, assuming that PS, 5-ALA-induced PpIX is activated at a wavelength of 635 nm. The availability of a general and high-quality treatment planning system renders iPDT more effective and applicable to a wider range of cancers [103].

The Monte Carlo method was employed to simulate light propagation in a glioblastoma brain model during iPDT and to evaluate treatment outcomes. The simulation evaluated the effects of fluence rate, tissue PS concentration, tissue oxygen saturation, and temperature on the PDT outcome, which was estimated based on the concentration of singlet oxygen produced. The results of the simulation demonstrated that the current standard study protocol (Es = 200 J/cm^2^), at an initial PS concentration of 5 μM, achieved a satisfactory level of glioblastoma cell killing in a relatively short treatment time while maintaining brain tissue temperature within an acceptable range. Furthermore, the simulation indicated that 39% of post-resection glioblastoma cells were killed by the treatment. Increasing the treatment duration and radiation fluence rate (E_s_ = 400 J/cm^2^) resulted in further cell killing, but also in an increase in tissue temperature, which may affect the safety of the treatment. An increase in PS concentration led to a significant increase in cell death: 61% of glioblastoma cells were killed when the PS concentration was doubled to 10 μM and the time and fluence rate were maintained [104]. 

In a further study, Monte Carlo simulation was employed to estimate the fluence rate of light in tumor tissue, with the objective of improving the light dosimetry of PDT. By simulating the dynamics of the PDT process, the spatial and temporal distributions of PS, ground state oxygen, and accumulated reactive singlet oxygen during treatment at different tumor depths were obtained. The qualitative behavior of these distributions was found to agree with experimental results published in the literature. A novel dosimetric quantity, tumor reactive singlet oxygen (TRSO), representing the amount of singlet oxygen per tumor volume that reacts with molecules in the tumor during treatment, is introduced to calculate the minimum radiation fluence required to destroy the tumor. It is demonstrated that the minimum radiation fluence required to destroy a tumor is nonlinearly dependent on tumor size. For tumors of a small size, the minimum fluence required to reach the optimal TRSO threshold is almost independent of tumor size. However, for tumors of a given critical size, this threshold is reached exponentially. This suggests that in clinical practice, the fluence setting is not a significant factor if the tumor is thin enough. Conversely, if the tumor is larger than the maximum size at which the TRSO threshold is reached, PDT is rendered ineffective [105].

Monte Carlo simulations were employed in conjunction with computed tomography images of 60 patients to investigate light delivery to the abscess cavity, assess the influence of optical properties and abscess morphology on the feasibility of PDT, and develop treatment plans. It was demonstrated that threshold optical power is dependent on abscess wall absorption and intralipid scattering within the cavity. The incorporation of optical properties into treatment planning significantly increased the number of patients who received an effective light dose of antimicrobial PDT, particularly in the context of intralipid modification [106].

A computational approach to personalized planning of iPDT for malignant central airway obstruction was presented. This approach simultaneously optimized fluence rate and fluence. This was achieved using the finite element method of Comsol Multiphysics 6.2 or the Dosie 0.0.3 software to study light propagation in biotissue (Figure 11) [107].

The outcomes of finite element modeling were validated by the findings of experimental light dosimetry in a solid phantom with optical properties corresponding to those of biotissues. The correlation coefficient between the simulation results in the Comsol and Dosie packages and the clinical imaging results of four patients with malignant central airway obstruction demonstrated a high degree of agreement in terms of radiation fluence rate and fluence [107]. 

The Monte Carlo method was employed to simulate the propagation of excitation photons and PpIX fluorescence photons in a three-dimensional model of normal skin tissue, including a tumor containing PS, i.e., PpIX, with the corresponding optical properties of normal and tumor tissue. The entire upper surface of the model was irradiated uniformly using the following parameters: a radiation wavelength of 632 nm, a fluence rate of 0.82 mW/cm^2^, and an irradiation time of 30 min, resulting in a radiation fluence of 150 J/cm^2^. The dosimetric parameter was defined as the number of photons absorbed by PS in a unit volume of tumor tissue. It was considered proportional to the local singlet oxygen yield. The threshold photodynamic dose resulted in tissue necrosis if the number of photons absorbed by PS in a unit volume of tumor tissue exceeded 8.6 × 10^17^ photons/cm^3^. The clinical results indicated that the dose constant for 5-ALA-induced PpIX photobleaching β was 14 ± 1 J/cm^2^ at a fluence rate of 82 mW/cm^2^. The change in surface fluorescence of PS during PDT due to photobleaching was employed to predict the depth of necrosis. Consequently, the necrosis depths by singlet oxygen generation in the tumor were 2.0, 2.7, 3.0, and 3.3 mm following the delivery of energy doses of 37.5, 75, 112.5, and 150 J/cm^2^, for a specific set of optical properties. The results demonstrated that an increase in the standard treatment light dose from 75 to 150 J/cm^2^ could enhance PDT efficiency at depths in tumors from 2.7 to 3.3 mm, respectively. Furthermore, this increase resulted in a reduction in the surface fluorescence of PpIX from 0.00012 to 0.000003 of the maximum value recorded prior to therapy. The recommendation to administer a higher dose of light, which involves increasing the treatment time after the reduction in PpIX surface fluorescence, remains relevant to the different optical properties of the tissues and should therefore have a positive result with respect to the overall treatment effect [97].

A straightforward and practical model was proposed for use with a spectral imaging system employed in real time to monitor hemoglobin oxygen saturation in human tissues during PDT. This model was based on a linear approximation of the dependence of the diffuse reflectance on the ratio of optical density to the reduced scattering coefficient. This dependence was obtained through Monte Carlo modeling of photon propagation in turbid media. The spectra of oxygenated and deoxygenated forms of hemoglobin differ mainly in the red region (520–600 nm) and have several characteristic points. Therefore, four band-pass filters were employed to obtain a multispectral image. Following the measurement of the reflection coefficient, the obtained data were utilized to select the concentration of oxygenated and free hemoglobin, as well as hemoglobin oxygen saturation [108].

The Monte Carlo method, as traditionally implemented, is highly computationally time-consuming. Consequently, several improvements have been developed with the objective of accelerating the modeling process. One such improvement is the use of graphics processing units (GPUs) to parallelize the algorithms, which represents a cost-effective solution for speeding up the Monte Carlo method. The parallel implementation of Monte Carlo using GPU technology was applied to PDT dosimetry using optical fibers with cylindrical diffusers. The outcomes of Monte Carlo simulations were then contrasted with experimental measurements using a homogeneous optical phantom that simulates the optical properties of brain tissue. The illumination profiles of five distinct cylindrical diffusers were quantified and compared with the Monte Carlo outcomes. Furthermore, this Monte Carlo algorithm was employed to corroborate ex vivo experimental measurements by evaluating the dosimetry of an illumination device designed for iPDT treatment of brain tumors (Figure 12) [109].

The outcomes of Monte Carlo verification with GPUs are in accordance with the recommendations of the American Association of Physicists in Medicine. The implementation speed of the Monte Carlo method using a GPU was found to be approximately 760 times faster than that of a central processor. The acceleration of the modeling process due to the GPU permitted its integration into the planning system for clinical iPDT [109].

In order to facilitate the use of open-source software in medical applications that perform the calculations required to plan treatment outcomes prior to in vivo PDT, the user-friendly, open-source, web-based FullMonteWeb 0.0.1 software with a graphical user interface for modeling, visualizing, and optimizing iPDT has been developed. The software facilitates Monte Carlo simulation of light propagation in biotissues without the necessity for complex setup. The open-source FullMonte package offers a more straightforward and cost-effective approach to interstitial PDT studies [110].

The development of parallel algorithms that have reduced computational time has demonstrated the relevance of replacing analytical modeling with graphically accelerated Monte Carlo simulation in the case of glioblastoma treatment. Nevertheless, analytical modeling may still be employed in optimization algorithms that can be utilized in a photodynamic treatment planning system. In a study, two analytical methods were evaluated to investigate the propagation of 635 nm laser radiation emitted from a cylindrical fiber source in a glioblastoma tissue model containing PpIX. The methods were distinguished by discrete or continuous modeling of the cylindrical diffuser. The discrete method entails the discretization of the scattering part of the optical fiber and its treatment as a sum of several point light sources. In contrast, the continuous method treats the fiber as a linear light source. The discrete method demonstrated superior correlation of the results of the estimation of the fluence rate of light in the tissue with those obtained by the Monte Carlo method in comparison to the continuous method. The means of all mean relative deviations of the values of light fluence rate were 1.23% (2.48%) for the discrete method and 26.12% (7.53%) for the continuous method. Statistical analysis (Student’s criterion) of all data sets confirmed that the two mean values were significantly different (*p* < 0.0001). With regard to the sensitivity study, the reduced scattering coefficient was found to be the most influential parameter on the variance in both analytical models in the near field of the light source. Conversely, at a distance greater than 3 mm from the light source, the absorption coefficient was identified as the most influential parameter in both analytical models [111]. 

Prognostic algorithms permit the optimization of treatment regimens, minimizing and quantifying the risk of spinal cord injury based on each patient’s anatomy and tumor volume. This is of critical importance for the effective application of PDT in clinical practice in this area. A platform for PDT modeling and planning was presented, and the feasibility of using BPD-MA-mediated PDT to treat bone metastases at two different wavelengths (690 nm and 565 nm) was investigated. The open-source PDT planning software, PDT-SPACE [112], was employed to assess the efficacy of distinct configurations of light-scattering fibers (end and cylindrical) with optimized power distribution. This was undertaken with the objective of minimizing the risk of spinal cord damage or maximizing the efficacy of tumor destruction. Three patients with spinal metastases were imaged using computed tomography for the purpose of planning PDT treatment. The simulation results indicated that PDT with a wavelength of 565 nm could potentially cure 90% of the metastatic lesion with less than 17% spinal cord damage. However, the required energy, and consequently, treatment time, to achieve this result at 565 nm wavelength is unattainable. The requisite energy and treatment time for the longer wavelength of 690 nm are attainable, but treatment targeting 90% of the metastatic lesion could result in significant damage to the proximal spinal cord. PDT-SPACE has provided a modeling platform that can be utilized to optimize PDT of metastatic spinal lesions [113].

Neural networks are increasingly being employed in place of Monte Carlo simulations, due to their ease of use and the significant time savings they offer over Monte Carlo simulations with comparable accuracy. A machine learning method called transfer learning was employed to perform a calibration between the diffuse reflectance coefficient modeled by Monte Carlo simulation and the coefficient measured experimentally by diffuse reflectance spectroscopy. The transfer learning model is capable of rapidly predicting the diffuse reflectance spectrum by training the model using a limited amount of diffuse reflectance spectroscopy data [114].

## 4. Conclusions

PDT has gained considerable traction in recent decades. This treatment modality employs light to induce the activation of PSs, which then mediate the destruction of cancer cells through the generation of ROS. The selectivity, minimal invasiveness, lack of side effects, and reduced toxicity and drug resistance inherent to PDT have led to its widespread adoption as a cancer treatment alternative to conventional modalities, such as chemotherapy and radiation therapy, that have been associated with adverse effects and drug resistance. Despite the widespread clinical application of this method, there is a pressing need for the development of an accurate, reliable, and straightforward method of dosimetry during PDT. The effective dose of light for a specific tumor is a crucial determinant of the success of the treatment, and the ability to determine this accurately is essential. The measurement of PDT dosimetry is challenging due to the necessity of simultaneously monitoring light dose, PS, and tissue oxygen concentrations in real time in order to fully characterize the biological response. The dosimetry methods presented in the article can be employed for the real-time monitoring and control of the PDT processing. These methods permit the visualization of PS distribution in tissues, as well as the measurement of its concentration and the assessment of the degree of oxygen saturation of tissues. They also permit the assessment of the optical properties of tumor tissue, which is of critical importance for the monitoring of tumor treatment effectiveness and the minimization of side effects. The practical challenges associated with the implementation of real-time PDT dosimetry techniques in clinical settings include the high cost of equipment and the need for regular maintenance, the need for staff training, and the development of clear protocols for using dosimetry data to optimize treatment. It is similarly crucial to take into account the technical difficulties inherent to the calibration and configuration of equipment, in addition to the interpretation of the data obtained.

The implementation of advanced PDT dosimetry techniques into clinical practice has the potential to result in improved quality of care and reduced overall healthcare costs in the long term. However, this will require a significant upfront investment and organizational effort. In essence, the use of these methods can enhance precision and treatment efficacy, potentially eliminating the need for supplementary interventions and shortening the recovery period for patients. Consequently, overall treatment costs can be reduced, and patient quality of life can be improved. Conversely, implementing novel PDT technologies requires expenses for personnel training, equipment acquisition, and protocol development. Additionally, there may be costs associated with maintenance and upgrades of equipment over time.

It is anticipated that the overall financial burden associated with cancer treatment will increase in the near future. At present, approximately half of the global economic costs associated with cancer are attributable to the treatment of specific types of the disease. These include cancer of the trachea, bronchus, and lungs (15.4%), colon and rectal cancer (10.9%), breast cancer (7.7%), liver cancer (6.5%), and leukemia (6.3%) [115]. Investment in research and development of cancer screening, diagnostics, and treatment has the potential to yield substantial health and economic benefits, particularly in low- and middle-income countries where cancer survival rates are comparatively lower than in high-income countries [116].

A multitude of disparate PDT dosimetry techniques have been described in the literature, though they share a common disadvantage: the inability to provide a comprehensive map of the effective light dose across the entire tumor tissue volume. This limitation is due to their inability to simultaneously estimate both PS concentration and hemoglobin oxygenation, either at a specific point in the tissue or over an averaged volume at a shallow depth. Table 1 presents the main advantages and limitations of current PDT dosimetry methods.

Consequently, there currently exists a surge of interest in the field of real-time dosimetry for PDT, with the aim of identifying alternative methods. 

Fluorescence imaging is widely used in clinical oncology for photodynamic diagnostics and dosimetry [117]. Previously, the possibility of Photofrin-mediated PDT of pleural mesothelioma was shown in 20 patients with simultaneous monitoring of the fluence rate of the treatment light and Photofrin fluorescence [58]. Multispectral luminescent dosimetry of singlet oxygen and PS photobleaching was applied to Verteporfin-PDT [118,119]. The authors in [120] demonstrated the feasibility of successful overt ROS dosimetry in Photofrin-mediated pleural PDT. In [121], in the treatment of 29 patients with moderately/well-differentiated microinvasive and squamous cell carcinomas of the oral cavity, the possibility of effective monitoring of ALA-PDT by changes in PpIX fluorescence intensity using a smartphone device was demonstrated. PpIX photobleaching dosimetry is also often used in ALA-PDT of brain tumors [50,122,123]. The authors in [48] demonstrated the effectiveness of dual-wavelength ALA-PDT dosimetry of patients with actinic keratosis (n = 15) using PpIX photobleaching with fluorescence excitation at 405 and 639 nm, which makes it possible to take into account the PS distribution in the superficial and deeper layers of biological tissues. Previously, the effectiveness of two-wave Ce6-PDT monitoring for changes in the PS fluorescence intensity at different depths of biological tissue was also demonstrated and the results were shown to correspond to the results of Monte Carlo simulation [124]. Currently, preliminary numerical modeling methods are also being introduced into clinical practice to predict optimal PDT energy doses. The authors in [106] demonstrated that individual planning of treatment of deep abscess cavities (n = 60) using Monte Carlo simulation allows the selection of an effective light dose of antimicrobial PDT.

The current limitations of existing techniques for assessing the entire tumor volume at great depth represent a significant challenge in the clinical practice of photodynamic tumor treatment. These limitations can have a considerable impact on the accuracy of photodynamic diagnosis and PDT efficacy. Incomplete assessment of tumor volume can result in an underestimation of its size, which in turn can lead to insufficient or excessive irradiation of tissues. This can subsequently lead to relapses after treatment or to superficial thermal damage to tissue. The specific depth ranges in which existing methods may be insufficient depend on the tumor type, its location, optical parameters, the PS type and its tissue distribution, and the energy parameters of the PDT. The depth of light penetration into tissues can be increased, thereby enabling the treatment of tumors located at great depths. This can be achieved by using light in the first (650–900 nm) or second (1050–1350 nm) “biological windows” for photodynamic diagnostics and therapy. However, the shift in the radiation used to the NIR region reveals new problems. First, the widely used cheap silicon detectors have virtually no sensitivity to wavelengths above 900 nm. The use of InGaAs detectors is necessary, although this entails a significant increase in cost. Secondly, the generation of singlet oxygen is optimized when a PS is excited at a wavelength of no more than 800 nm. Otherwise, the efficiency of singlet oxygen generation is significantly reduced [125]. Despite the advancements that have been made in this field, there are still limitations associated with the depth of light penetration and the accuracy of dosimetry. Further research in this area is ongoing, with the expectation that future developments will enhance the accuracy and efficacy of PDT for the treatment of deep-seated tumors.

The visible and near-infrared spectral range is typically used as a source for PDT. Nevertheless, in the presence of an appropriate photosensitizing agent, the optimal sources for PDT are those emitting within the near-infrared spectral range, between wavelengths of 700 and 1000 nm. This wavelength range is characterized by minimal absorption and scattering of light by biological tissues. This feature is beneficial for both optical imaging and phototherapy. In this context, a novel optical approach for continuous and real-time monitoring of absorbed near-infrared laser radiation dose within aqueous solutions, which is regarded as the optimal medium for tissue-equivalent measurements due to its high water content (accounting for the majority of the human body weight), is proposed, and its underlying principles as well as its limitations are fully outlined. As a consequence of the aforementioned research, profiles of light penetration and the dose distribution have been determined. Additionally, a relationship between measurements obtained using the presented method and dose value has been established [126].

The integration of diagnostics and therapeutics, designated as theranostics, into a unified system has long been regarded as a strategy for achieving personalized and precise tumor treatment. Nevertheless, the successful implementation of tumor theranostics necessitates the careful selection of compatible tumor diagnostic and treatment modalities and their effective combination into a single system for clinical application. Moreover, the combination of fluorescence imaging and PDT is currently under investigation for the purpose of tumor theranostics. Raman spectroscopy is an optical diagnostic technique that employs the application of laser light and subsequent collection of inelastically scattered light from the tissue to identify subtle biochemical differences between tissues. For in vitro cell theranostics and in vivo mouse tissues, a 2 mm diameter multimodal fiber optic probe was utilized. This probe consisted of a central fiber for Raman spectroscopic excitation, and seven optical fibers arranged around the central fiber. Two of these optical fibers were used for PDT at different wavelengths, while the remaining five were employed for the collection of Raman spectroscopic light scattering signals. It has been demonstrated that by selecting an appropriate excitation wavelength for Raman spectroscopy and carefully selecting the PS for PDT, it is possible to achieve effective tumor theranostics without interaction between the two methods. The fluorescence of the investigated PSs did not affect the Raman spectral information obtained. The high fluorescence yield of the PS and the low background Raman fluorescence of the PS allowed the tumors to be visualized in real time during PDT, and the outcomes of the treatment were also predicted [127,128]. In vivo studies were also conducted in rats with mammary tumors, in which combined Raman spectroscopy and PDT were employed. This approach allowed for the classification of spectra before and after therapy with 100% efficiency [129].

Optical coherence tomography (OCT) represents a highly promising optical imaging method, offering clinicians timely and accurate insights into the tissue response to PDT treatment. This, in turn, enables them to optimize therapy parameters with a view to improving the prognosis for patients and minimizing adverse treatment outcomes. Furthermore, the OCT method is capable of detecting submillimeter-sized tumor nodules, thereby enabling the detection of residual tumors. The OCT method was employed to ascertain the precise boundaries of patients’ precancerous and cancerous skin lesions and to monitor skin changes subsequent to PDT. A correlation was established between the measured OCT parameters and the treatment outcome, thereby allowing for real-time adjustment of tissue irradiation parameters during PDT. This resulted in a notable enhancement in treatment efficiency. The enhanced visualization of skin neoplasm boundaries may necessitate the irradiation of a smaller area, thereby preserving a greater proportion of normal tissue. The OCT images revealed distinct microstructural differentiations between normal skin, precancerous skin, cancerous skin, and the transition zone between the two tissues. The observed improvement in cosmetic outcome was rated as “excellent” by 89% of patients [130].

A further approach to optimizing the efficacy of vascular-targeted photodynamic therapy (V-PDT) involves the accurate assessment of tumor vascular network changes in response to the treatment. Nevertheless, accurate assessment of tumor vasculature changes, particularly microvascular alterations in vivo, remains a challenging task due to limitations inherent in existing imaging modalities in terms of spatial resolution. OCT in the visible spectrum has demonstrated the potential for non-contact oximetry and high-resolution vascular visualization [131,132], due to the restricted depth of penetration of visible light into biological tissues. Studies have demonstrated the capacity of M-mode OCT operating at a wavelength of 1300 nm to detect V-PDT-induced blood vessel changes in a mouse ear tumor model [133,134]. However, the suboptimal resolution of ~10 μm OCT at the 1300 nm wavelength makes it challenging to visualize the microcirculatory channel, particularly in tumors with a diameter of approximately 7 μm [135]. Conversely, operating the OCT at a wavelength of 800 nm affords a resolution of less than 3 μm, which could provide an unparalleled opportunity to assess the response of the tumor vascular network to V-PDT. In vivo imaging of murine tumor microvessels during V-PDT with verteporfin was demonstrated using an ultra-high resolution OCT system operating at a wavelength of 800 nm. Newly formed blood vessels with a diameter of approximately 10 μm in the tumor could be readily identified. V-PDT led to a pronounced reduction in the diameter of tumor veins and vessels, with only a minor impact on arterial dimensions. The real-time in vivo OCT imaging and quantification of the tumor microcirculatory response to V-PDT in vivo could potentially enhance our understanding of V-PDT processes and facilitate the achievement of optimal tumor treatment outcomes in the future [136].

To obtain a visual representation of the dynamics of blood flow and vascular structure in vivo, with both spatial and temporal resolution, a non-invasive methodology based on laser speckle contrast imaging (LSCI) is employed. This technique is founded upon the theoretical principle of coherent laser radiation, which illuminates tissue, thereby causing backscattered light to form a random interference pattern on a detector. This phenomenon is referred to as a speckle pattern [137]. LSCI has been employed extensively due to its advantages of high speed, wide field of view, and ease of implementation, with a particular focus on the visualization of blood flow and vascular damage during V-PDT [138]. The method was employed to visualize microcirculation and to monitor microcirculatory alterations in port wine stains (PWSs) during V-PDT, which involved the intravenous administration of photocarcinorin (4–5 mg/kg). Prior to undergoing V-PDT, all 24 patients with port wine stains demonstrated a statistically significant difference in perfusion between the lesion and the contralateral, healthy skin (1132 ± 724 and 619 ± 478 perfusion units (PU), respectively; *p* < 0.01). Five minutes after V-PDT, the mean perfusion units of the 24 PWS foci were 1246 ± 754 PU. There was no statistically significant difference compared to the perfusion observed prior to V-PDT (*p* > 0.05). During V-PDT, the perfusion of the seven PWS patients exhibited a rapid increase following the commencement of the procedure, reaching a peak within 10 min. This continued for several minutes before gradually returning to a relatively lower level at the conclusion of V-PDT [139]. Laser speckle imaging (LSI) was employed to investigate 17 patients with 40 PWS lesions before and between three and six months after V-PDT. A correlation was established between perfusion changes and photobleaching. Prior to V-PDT, the perfusion of 40 PWS foci was observed to be higher than that of normal skin (1421 ± 463 and 1115 ± 386 PU, respectively; *p* < 0.01). The follow-up scans of the PWS lesions demonstrated a reduction in perfusion compared to the values observed prior to V-PDT (1282 ± 460 and 1421 ± 463 PU, respectively; *p* < 0.01). Following V-PDT, the rate of change in perfusion was found to be in close alignment with the rate of photobleaching (correlation coefficient of 0.73) [140]. These findings demonstrated that LSI could effectively visualize the PWS microvascular network and monitor the microvascular response to V-PDT [139]. LSI was also utilized to assess the blood flow dynamics in PWS during combined PDT and pulsed dye laser exposure, resulting in a synergistic effect that facilitated persistent vessel closure. It is hypothesized that the use of acute photocoagulation and persistent vessel closure may be key factors leading to improvements in outcomes for the treatment of PWS. Further integration of non-invasive optical imaging technologies and biochemical analysis may result in the development of more effective and reliable treatment strategies [141]. The use of dual-wavelength reflectance imaging has been shown to improve the visualization of laser speckles of blood vessels, thereby increasing the signal-to-noise ratio during V-PDT. In vivo studies have demonstrated that the proposed method enhances the signal-to-noise ratio by 4–29%, rendering it an effective instrument for monitoring blood flow and vascular injury during V-PDT.

Currently, there is a significant focus on the advancement of intraoperative PDT dosimetry methods, with the objective of identifying the optimal parameters for light irradiation in a personalized treatment approach. It is well established that tumors exhibit considerable heterogeneity, which may impede the efficacy of photodynamic diagnostics and therapy, especially in deep-seated tumors. One potential solution to this issue may be the preliminary prediction of light propagation throughout the entire tumor volume for different irradiation geometries and for different configurations of light-scattering fibers [60]. The utilization of a three-dimensional tumor model, previously obtained through computed tomography or magnetic resonance imaging (MRI) with Monte Carlo simulation, permits the simulation of PDT within the tumor volume [104], and the walls of hollow organs [100,106]. An optical scanning method represents a low-cost and effective alternative to CT and MRI for the construction of a model of organ walls [142,143]. For example, the results of PDT modeling using an individual 3D model of metastatically affected vertebrae obtained from three patients demonstrated the potential for the treatment of 90% of metastatic lesions with less than 17% spinal cord damage [113]. The integration of machine learning algorithms in the PDT planning system enables the prediction of changes in tissue optical properties with an error rate of less than 6%, thereby enhancing the reliability of the predicted PDT outcome by significantly reducing the discrepancy between the predicted and ideal plan outcomes [144]. These methods can be combined with other PDT dosimetry methods, such as explicit dosimetry of PS photobleaching or ROS generation, which will markedly enhance the reliability of the predicted result of PDT and will further augment the efficacy of treatment of deep-seated tumors.

## Figures and Tables

**Figure 1 cancers-16-02484-f001:**
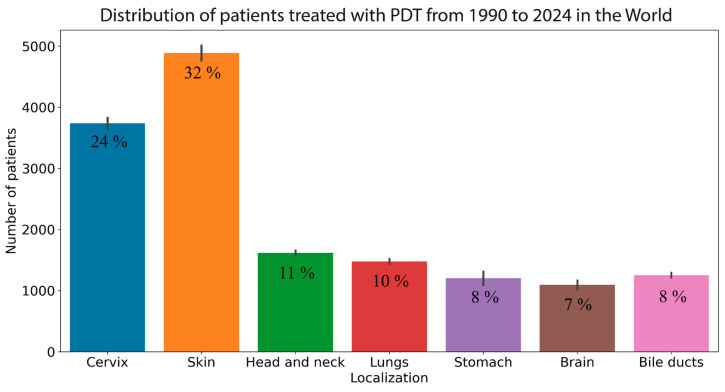
A distribution of patients treated with PDT by localization from 1990 to 2024 across the globe.

**Figure 2 cancers-16-02484-f002:**
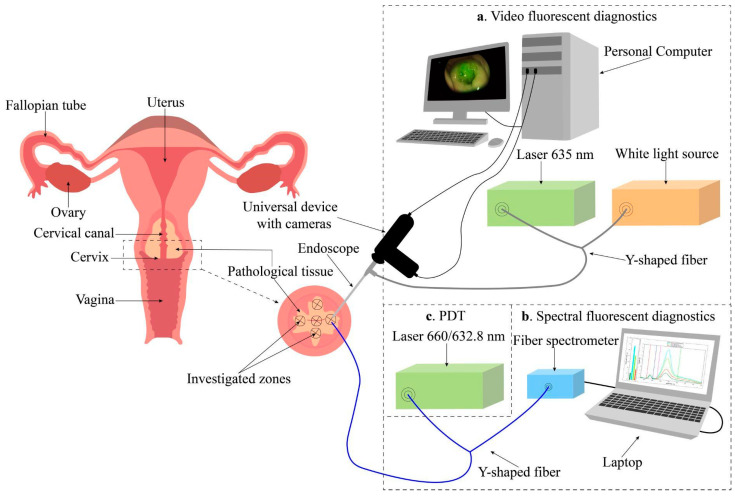
A schematic diagram of a cervical neoplasm boundary examination by means of spectral and video fluorescence diagnostic methods [52]. Reproduced with permission from IOP Publishing.

**Figure 3 cancers-16-02484-f003:**
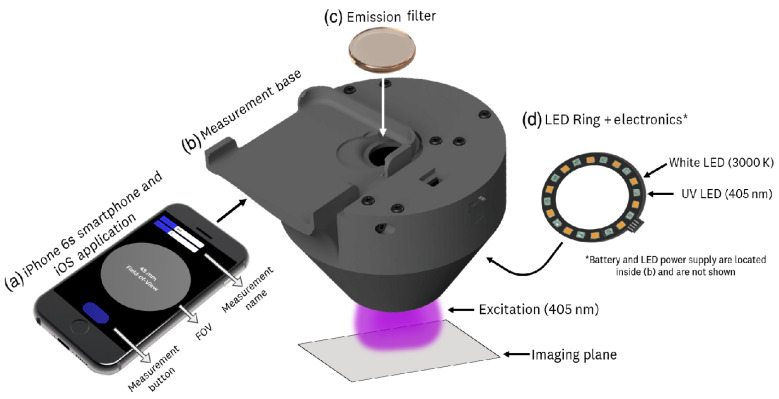
Schematic diagram of the dosimetry system [56]. Reproduced with permission from SPIE.

**Figure 4 cancers-16-02484-f004:**
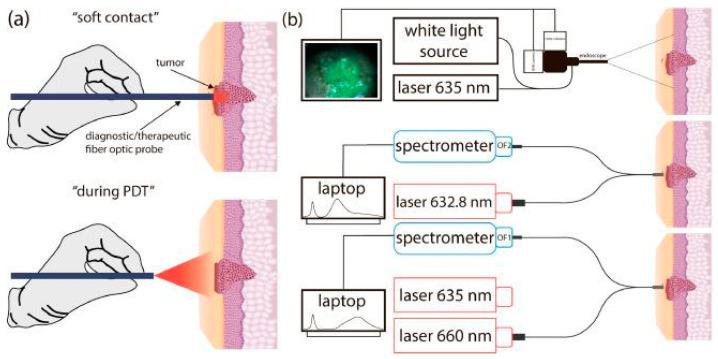
Tumor phototheranostics scheme: (**a**) “soft contact” and “during PDT” modes; (**b**) video- and spectral-fluorescence diagnostics, phototheranostics with 635 or 660 nm lasers [16]. Reproduced with permission from Elsevier.

**Figure 5 cancers-16-02484-f005:**
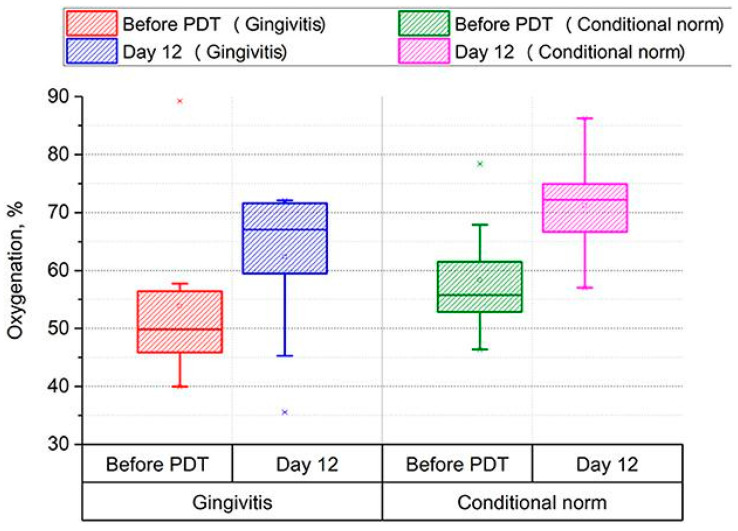
Assessment of gingival tissue oxygenation level before and after PDT (*p* < 0.001, power > 0.99) [68]. Reproduced with permission from Frontiers.

**Figure 6 cancers-16-02484-f006:**
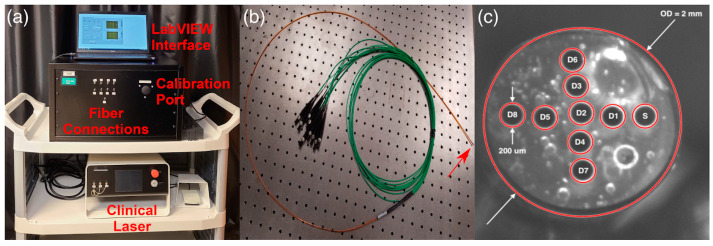
Scheme of the study. (**a**) Spectroscopy system; (**b**) optical fiber with distal end indicated by red arrow; (**c**) cross-section of the distal end of the fiber [70]. Reproduced with permission from SPIE.

**Figure 7 cancers-16-02484-f007:**
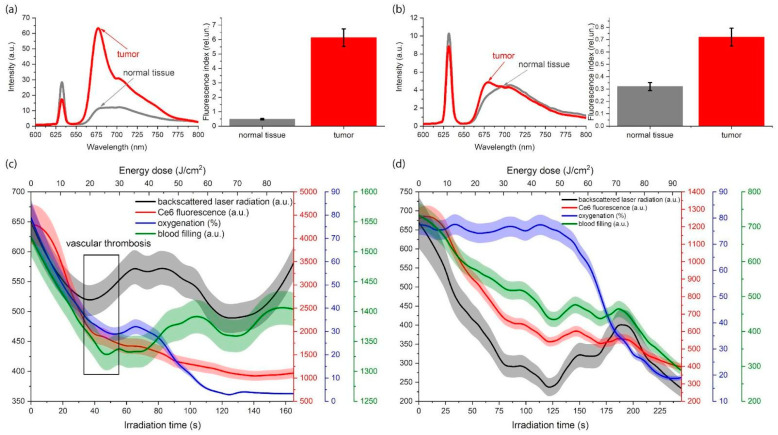
Combined monitoring of photodynamic irradiation: (**a**,**b**) spectral-fluorescence diagnostics before PDT; (**c**,**d**) distributions of integral intensities of diffusely scattered laser radiation, Ce6 fluorescence, and hemoglobin oxygenation during PDT [78]. Reproduced with permission from Elsevier.

**Figure 8 cancers-16-02484-f008:**
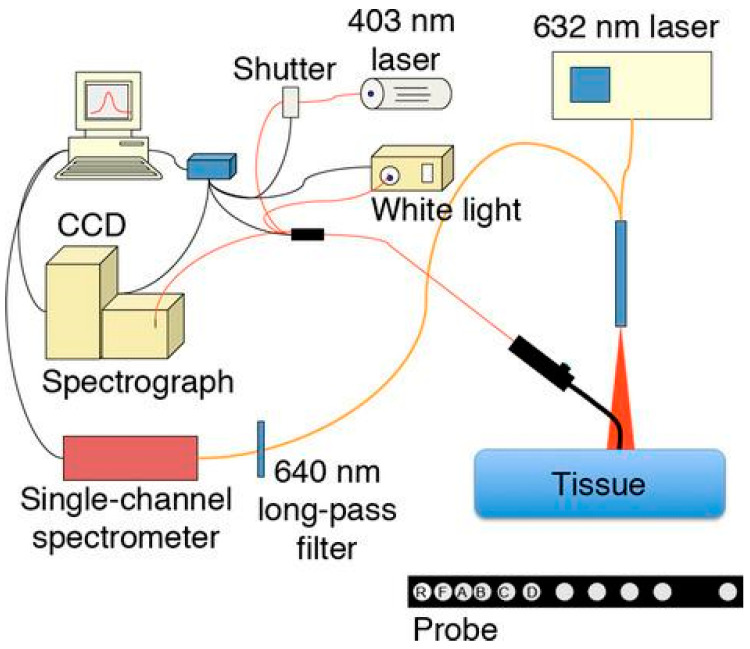
Scheme of the system for diffuse reflectance/fluorescence spectroscopy. The probe profile is shown in the lower right. The reflectance source (R), fluorescence source (F) and the first four detection fibers (A through D) are labeled [82]. Reproduced with permission from Wiley.

**Figure 9 cancers-16-02484-f009:**
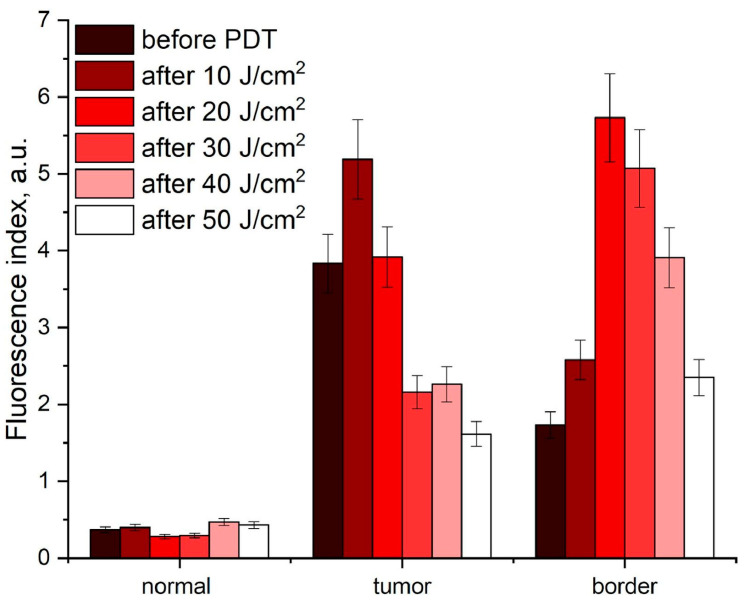
Distribution of fluorescence indices in tissues before and after low-dose PDT [83]. Reproduced with permission from Elsevier.

**Figure 10 cancers-16-02484-f010:**
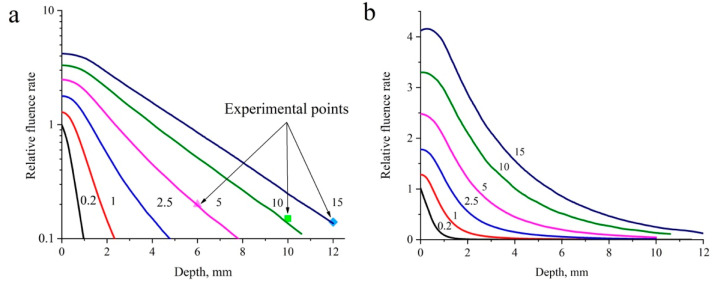
Modeled dependence of the relative fluence rate of laser radiation on the penetration depth at different spot diameters (indicated by the number near the curve; mm) in the cervical tissue. (**a**) Logarithmic scale; (**b**) linear scale [88]. Reproduced with permission from IOP Publishing.

**Figure 11 cancers-16-02484-f011:**
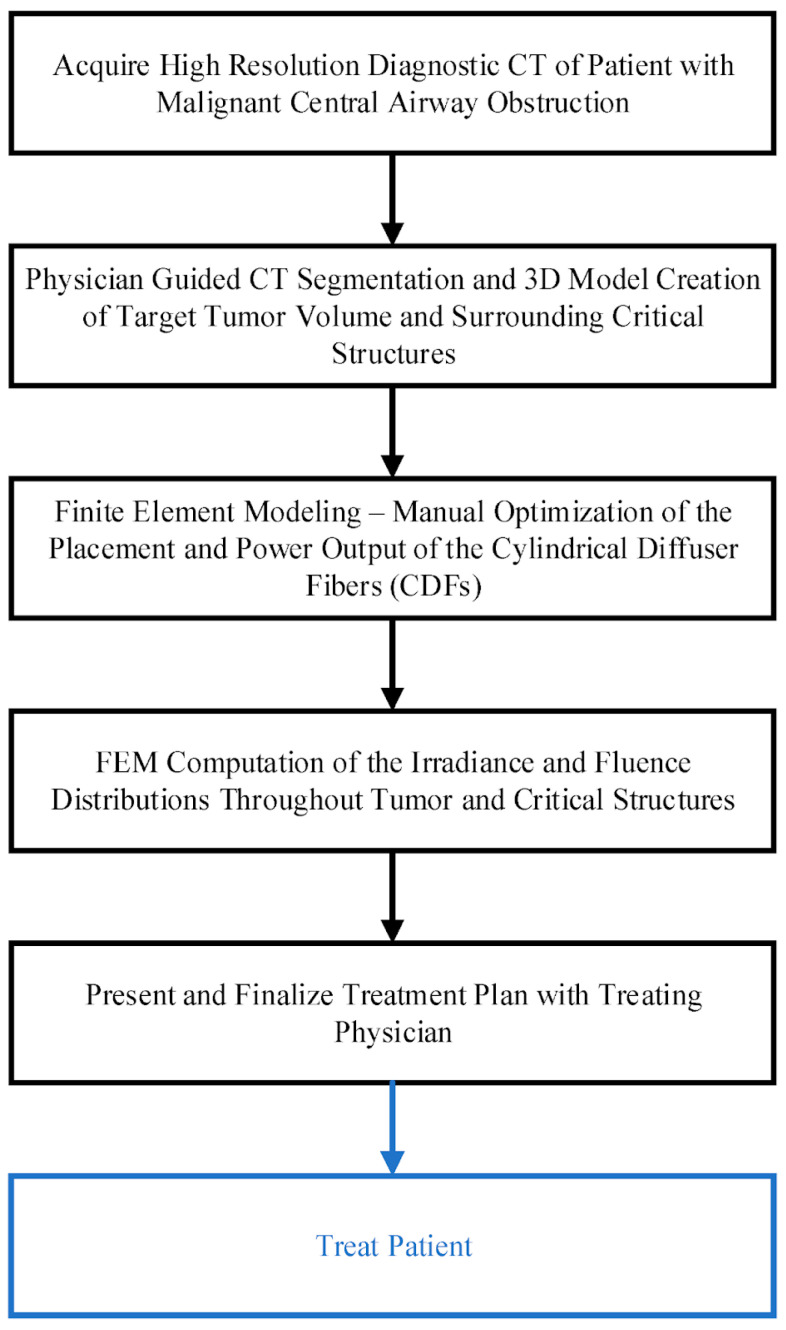
Diagram of the step-by-step treatment planning procedure for i-PDT [107]. Reproduced with permission from MDPI.

**Figure 12 cancers-16-02484-f012:**
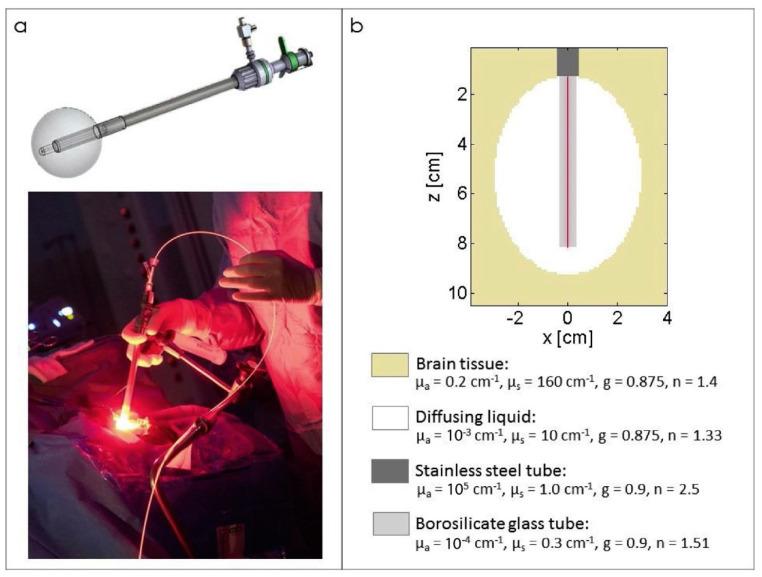
The iPDT treatment of brain tumors is a procedure that has been developed to improve the effectiveness of surgical removal of these tumors. (**a**) The light applicator comprises a fiber guide inserted into the balloon to guide the 70 mm cylindrical diffuser. This is a clinical application. (**b**) A model of the device filled with 150 mL of a diffusing solution was constructed. The 70 mm long cylindrical diffuser was positioned at the center of the borosilicate glass tube [109]. Reproduced with permission from Elsevier.

**Table 1 cancers-16-02484-t001:** Advantages and limitations of photodynamic dosimetry methods.

Method	Advantages	Limitations
Video fluorescence imaging	real-time mapping of interstitial PS distribution relative ease of implementation in operational settings possibility of visualizing tumor boundaries ability to perform dosimetry directly during the PDT process ability to study the dynamics of PS accumulation in tissues	difficulty in separating endogenous and exogenous fluorescence difficulty in monitoring changes in the photochemical properties of PS molecules during PDT difficulty monitoring blood filling and PS photobleaching simultaneously necessary to include additional light sources to obtain fluorescence signals from different tissue depths
Spectral fluorescence imaging	high spectral resolution possibility of spectral identification of endogenous and exogenous fluorescence ability to perform dosimetry during PDT ability to obtain fluorescence signals from different tissue depths by changing the measurement geometry (source–receiver) high signal-to-noise ratio	small depth of tissue probing volume obtaining information from a limited area/volume of tissues difficulty in assessing the absorption and scattering properties of tissues separately difficulty in mapping all sensitized pathological tissues
Diffuse reflectance spectroscopy	ability to simultaneously monitor tissue oxygen saturation and PS photobleaching monitoring the optical properties of tissues that change during PDT ability of combined monitoring of the PS photobleaching and blood filling during PDT	small depth of tissue probing volume obtaining information from a limited area/volume of tissues difficulty in monitoring optical properties of tissues in the absorption/fluorescence spectral range of PS
Computer simulation	ability to select optimal radiation modes for the entire tumor volume ability to predict changes in tissue optical properties during PDT PDT dosimetry with consideration of the irradiation geometry no influence on the results of external diagnostic radiation	dynamic changes in tissue optical properties during PDT dynamic changes in the photophysical properties of PS molecules as a function of the microenvironment dynamic changes in blood flow during PDT
Singlet oxygen dosimetry	direct dosimetry of ROS generation dosimetry directly during the PDT process	low signal-to-noise ratio for luminescence monitoring of singlet oxygen technical complexity of implementation under operational conditions
Raman spectroscopy	high spectral and spatial resolutions possibility of monitoring the concentration of non-fluorescent tissue components	difficulty in detecting low concentrations of substances high background PS fluorescence high laser power density technical difficulty of implementation in an operational environment difficulty in performing in vivo measurements
Optical coherence tomography	direct monitoring of the state of the tumor vasculature ability of monitoring saturation and blood flow high spatial resolution	applicable for PDT dosimetry only with vascular mechanisms of PDT action lack of monitoring of photochemical changes in PS molecules complexity of real-time dosimetry implementation technical complexity of implementation in operational conditions

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
