# Peer review of "Devices and Methods for Dosimetry of Personalized Photodynamic Therapy of Tumors: A Review on Recent Trends"

_cancers, 2024, doi:10.3390/cancers16132484_

Round 1

Reviewer 1 Report

Comments and Suggestions for Authors

The review is devoted to the consideration of diagnostic techniques and methods used to personalize photodynamic therapy of tumors. Separately, the authors dwell on the limitations of technology and methods of diagnostics and dosimetry. The review is written clearly and clearly, and is easy to read. In principle, I like to read reviews of groups from Vavilova Street on agricultural and medical-biological topics. There are many illustrations in the review; the main comments are related to them. I propose to make a number of minor improvements to the manuscript. 1. In Figure 1, I suggest that the authors write in numbers the percentage of the total on each column. 2. In Figure 2, some of the inscriptions are not possible to read, please enlarge them. 3. Figure 7 clearly does not have a sufficient description for understanding. It probably wouldn't hurt to expand the description. 4. In Figure 10 there is no interpretation of the curves shown, without this it is impossible to understand what is shown on the graph. 5. I propose to make a list of abbreviations. There are quite a lot of them in the text of the manuscript. Some are not deciphered at all, for example, the abbreviation ROS appears twice in the text. I believe that the manuscript can be published after making these changes; there is no need to send the manuscript to me for additional review.

Author Response

Comments 1[In Figure 1, I suggest that the authors write in numbers the percentage of the total on each column.]

Response 1 [We have modified Figure 1 and added it to the manuscript.]

Comments 2[In Figure 2, some of the inscriptions are not possible to read, please enlarge them.]

Response 2 [We have enlarged the captions for Figure 2 and added it to the manuscript.]

Comments 3[Figure 7 clearly does not have a sufficient description for understanding. It probably wouldn't hurt to expand the description. ]

Response 3 [We have expanded the description for Figure 7.]

Comments 4[In Figure 10 there is no interpretation of the curves shown, without this it is impossible to understand what is shown on the graph.]

Response 4 [We have added the interpretation of the curves.]

Comments 5[I propose to make a list of abbreviations. There are quite a lot of them in the text of the manuscript. Some are not deciphered at all, for example, the abbreviation ROS appears twice in the text.]

Response 5 [We have made a list of abbreviations.]

Reviewer 2 Report

Comments and Suggestions for Authors

In the manuscript, the authors summarize indicators and assays for personalized assessment of the therapeutic adequacy of photodynamic therapy (PDT) in the clinic, and outline them from different perspectives, with a focus on their use in vivo treatment and prognosis of tumors. Using in vivo photosensitizer (PS) consumption, hemoglobin oxygenation and tissue refractive index, this paper introduces the latest development of personalized tumor PDT latent dose control by means of optical imaging and computer modeling. The authors summarized the following advantages of personalized dose control of PDT: 1. For patients with different conditions, different degrees of light therapy can avoid residual PS accumulation caused by insufficient light dose and superficial heat damage of tissues caused by excessive dose. ; 2. The best measurement control through different detection means has the best therapeutic effect for patients in different periods; 3. The integration of diagnosis and treatment provides reference and development prospects for accurate tumor diagnosis and treatment. All in all, this review is a good summary of existing work and instructive. Publication is recommended after the following issues have been addressed:

1. In line 68 of the manuscript, it is mentioned that dominant dosing is used for non-overly invasive treatment, and then in the following examples it is mentioned that it is mainly used for overly invasive treatment. Is this a contradiction?

2. In line 110, is it mentioned that high and low flux light exposure can lead to oxygen depletion?

3. In line 437, how does the backscatter therapeutic radiation intensity affect Ce6-PDT? What does the change in this indicator show in the graph? The article only mentioned that it is an important indicator, and attached a graph.

4.There are some problems in the format of some references, such as the title of the article does not need to use italics, the hyperlink of the literature website is incomplete, and so on, which need to be further improved and modified.

Author Response

Comments 1 [In line 68 of the manuscript, it is mentioned that dominant dosing is used for non-overly invasive treatment, and then in the following examples it is mentioned that it is mainly used for overly invasive treatment. Is this a contradiction?]

Response 1 [We agree with this remark. We have changed the manuscript.]

Comments 2 [In line 110, is it mentioned that high and low flux light exposure can lead to oxygen depletion?]

Response 2 [We have made a clarification in the manuscript.]

Comments 3 [In line 437, how does the backscatter therapeutic radiation intensity affect Ce6-PDT? What does the change in this indicator show in the graph? The article only mentioned that it is an important indicator, and attached a graph.]

Response 3 [We have added a detailed description of this indicator to the manuscript.]

Comments 4 [There are some problems in the format of some references, such as the title of the article does not need to use italics, the hyperlink of the literature website is incomplete, and so on, which need to be further improved and modified.]

Response 4 [We have improved and modified references in the manuscript.]

Reviewer 3 Report

Comments and Suggestions for Authors

This is a quite full, detailed and comprehensive review of tumors PDT. the authors described recent trrends, demonstrated different approaches and highligted drawbacks of PDT systems presented in the field. So only some minor issues to improve the text:

1. The last sentece in the Summary is not finished: "... rendering it an effective instrument for monitoring blood flow and vascular injury during " .

2. The images in the text were taken from different papers, so it is advised to add phrases "reproduced with permission from..."

3. The sections Results and Discussion are missing.  Please, add them to the text structure.

4. It iadvised to add exact results to the Abstract (eg how many papers were analyzed; what are the most prominent wavelenghts for PDT, etc).

In general the paper leaves positive impression and may be published after correction of mentioned issues.

Author Response

Comments 1 [The last sentece in the Summary is not finished: "... rendering it an effective instrument for monitoring blood flow and vascular injury during "]

Response 1 [We have finished the last sentence.]

Comments 2 [The images in the text were taken from different papers, so it is advised to add phrases "reproduced with permission from..."]

Response 2 [We have added phrases "reproduced with permission from..."]

Comments 3 [The sections Results and Discussion are missing.  Please, add them to the text structure.]

Response 3 [We have added the "Results and discussion" section in the manuscript.]

Comments 4 [It iadvised to add exact results to the Abstract (eg how many papers were analyzed; what are the most prominent wavelenghts for PDT, etc).]

Response 4 [We have added exact results to the Abstract.]

Reviewer 4 Report

Comments and Suggestions for Authors

Alekseeva et al. extensively examines the current status and challenges of personalized PDT dosimetry for cancer treatments. It emphasizes the critical need to develop accurate dosimetric techniques to improve the efficacy of PDT and highlights fluorescence imaging, diffuse reflectance spectroscopy and computer modelling as promising methodologies. Considering the high quality of the manuscript, I would just like to make a few comments and questions to which I would like concrete answers and for the manuscript to be slightly improved: 1) The authors discusses various dosimetry methods, but could benefit from a more structured comparative analysis. A table summarizing the advantages and limitations of each method would provide a clearer understanding of their relative effectiveness and practical applications.

2) the article may benefit from the inclusion of more examples or case studies in which these dosimetry methods have been successfully applied would illustrate their benefits and practical challenges.
4) The authors mention the limitation of existing techniques in assessing the entire tumor volume at great depths. How important is this limitation in clinical practice and what are the specific depth ranges in which these methods are insufficient? are efforts being made to improve depth penetration in dosimetry?

5) to what extent do the reviewed dosimetry methods enable real-time monitoring of PDT? what are the practical challenges associated with implementing real-time dosimetry in clinical settings? Please comment.
6) The review emphasizes the need for personalized dosimetry. what patient-specific variables are most important for personalizing PDT and how are these variables currently measured or estimated?

7) What are the cost implications of adopting advanced dosimetry methods in clinical practice?

8) the manuscript calls for more research and innovation. what are the most promising technological innovations that could solve the current limitations of dosimetry? how close do you think are these innovations to clinical implementation?
9) finally, the review paper could explore how emerging technologies, such as machine learning, can be integrated into current dosimetry methods to improve prediction accuracy and personalized PDT.

Author Response

Comments 1 [The authors discusses various dosimetry methods, but could benefit from a more structured comparative analysis. A table summarizing the advantages and limitations of each method would provide a clearer understanding of their relative effectiveness and practical applications.]

Response 1 [We have made a table summarizing the advantages and limitations of each method and have added it to the manuscript.]

Comments 2 [the article may benefit from the inclusion of more examples or case studies in which these dosimetry methods have been successfully applied would illustrate their benefits and practical challenges.]

Response 2 [We have added more examples  in which these dosimetry methods have been successfully applied to the manuscript.]

Comments 4 [The authors mention the limitation of existing techniques in assessing the entire tumor volume at great depths. How important is this limitation in clinical practice and what are the specific depth ranges in which these methods are insufficient? are efforts being made to improve depth penetration in dosimetry?]

Response 4 [We have added the information to the manuscript.]

Comments 5 [to what extent do the reviewed dosimetry methods enable real-time monitoring of PDT? what are the practical challenges associated with implementing real-time dosimetry in clinical settings? Please comment.]

Response 5 [We have added a commentary to the manuscript.]

Comments 6 [The review emphasizes the need for personalized dosimetry. what patient-specific variables are most important for personalizing PDT and how are these variables currently measured or estimated?]

Response 6 [We have added patient-specific variables that are most important for personalizing PDT to the manuscript. These parameters can provide an integrated parameter that is closely correlated with the therapeutic dose. Currently, these parameters are mainly assessed using methods such as fluorescence imaging and diffuse reflectance spectroscopy.]

Comments 7 [What are the cost implications of adopting advanced dosimetry methods in clinical practice?]

Response 7 [We have added the information to the section “Summary and Perspective” of the manuscript.]

Comments 8 [the manuscript calls for more research and innovation. what are the most promising technological innovations that could solve the current limitations of dosimetry? how close do you think are these innovations to clinical implementation?]

Response 8 [We conducted additional research and have supplemented the section “Summary and Perspective” of the manuscript.]

Comments 9 [finally, the review paper could explore how emerging technologies, such as machine learning, can be integrated into current dosimetry methods to improve prediction accuracy and personalized PDT.]

Response 9 [We explored how emerging technologies, such as machine learning, can be integrated into current dosimetry methods and have updated the section “Summary and Perspective” of the manuscript.]

Round 2

Reviewer 2 Report

Comments and Suggestions for Authors

This manuscript have been improved after revision.